SciPost Physics

Submission

# Josephson effects between the Kitaev ladder superconductors

Osamu Kanehira[1] and Hiroki Tsuchiura[1,2]⋆

**1** Department of Applied Physics, Tohoku University, Sendai, Japan
**2** Center for Spintronics Research Network, Sendai, Japan
* tsuchi@tohoku.ac.jp

August 15, 2022

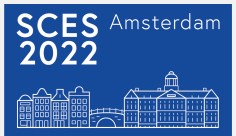

*International Conference on Strongly Correlated Electron Systems
(SCES 2022)
Amsterdam, 24-29 July 2022*

## Abstract

The two-leg ladder system consisting of the Kitaev chains is known to exhibit a richer phase diagram than that of the single chain. We theoretically investigate the variety of the Josephson effects between the ladder systems. We consider the Josephson phase difference $\theta$ between these two ladder systems as well as the phase difference $\phi$ between the parallel chains in each ladder system. The total energy of the junction at $T = 0$ is calculated by a numerical diagonalization method as functions of $\theta$, $\phi$, and also a transverse hopping $t_\perp$ in the ladders. We find that, by controlling $t_\perp$ and $\phi$, the junction exhibits not only the fractional Josephson effect for the phase difference $\theta$, but also the usual 0-junction and even $\pi$-junction properties.

# 1 Introduction

A topological Josephson junction consisting of two Kitaev superconducting chains exhibits a fractional Josephson effect with a $4\pi$ periodicity in the current-phase relationship [1]. This is a promising experimental proof for the existence of Majorana zero modes (MZMs) at the edges of the Kitaev chains. Experimentally, several methods have been proposed to realize the Kitaev chains, and a promising method is to create $p$-wave pairings by the proximity effect between an $s$-wave superconducting substrate with strong spin-orbit interaction and semiconductor nanowires [2]. Furthermore, theoretically, multi-terminal Josephson junctions of Kitaev chains have been proposed to realize the so-called braiding operation of MZMs to perform topological quantum calculations [3]. In such hybrid systems, the Majorana fermion signature may appear as a zero-bias peak in conductance, and many experimental attempts have been made [4,5]. However, no conclusive evidence for the existence of MZM, let alone braiding operations, has yet been obtained.

Despite the experimental difficulty of realization and the theoretical simplicity, the Kitaev chain is still attracting considerable interest. One direction is to study the ladder systems consisting of multiple Kitaev chains. It has been reported that a richer phase diagram and novel phenomena can be obtained even in minimal two-leg ladder systems [6–9]. The Hamiltonian of the two-leg Kitaev ladder system in general is given as

$$H = -\mu \sum_{i=1}^{N} \sum_{j=1,2} c_{i,j}^{\dagger} c_{i,j} - \sum_{i=1}^{N-1} \sum_{j=1,2} \left[ t\, c_{i+1,j}^{\dagger} c_{i,j} + \Delta_j c_{i+1,j}^{\dagger} c_{i,j}^{\dagger} + \text{h.c.} \right]$$
$$- \sum_{i=1}^{N} \left[ t_{\perp} c_{i,1}^{\dagger} c_{i,2} + \Delta_{\perp} c_{i,1}^{\dagger} c_{i,2}^{\dagger} + \text{h.c.} \right] \tag{1}$$

where $i = 1, 2, \cdots, N$ is the site number of the each chain and $j = 1, 2$ is the chain number in the Kitaev ladder. Also, $t$ is the nearest-neighbor transfar integral, $\mu$ is the chemical potential, $\Delta_j = \Delta e^{i\phi_j}$ is the intra-chain $p$-wave superconducting pair potential with phase $\phi_j$, and $t_{\perp}$ and $\Delta_{\perp}$ is the coupling between two chains. As is well known, in the single Kitaev chain, the topological phase is characterized by the $\mathbb{Z}_2$ invariant, and the phase diagram is described only by $\mu/t$. In contrast in the two-leg Kitaev ladder systems, a $\mathbb{Z}$ topological invariant characterizes the topological phase, and the inter-chain parameters $t_{\perp}$ and $\Delta_{\perp}$ are also affect the phase diagram. Recently, Maiellaro *et al.* has shown that the system exhibits a topological phase either with four or two Majorana zero-energy modes [7]. They also find that the topological phase survives also when the Kitaev's criterion $\Delta > 0$ and $|\mu| < 2t$ for the single chain is violated.

Because of the multiplicity of the Majorana zero modes and topological quantum numbers, we can naturally expect that the Josephson effect in the ladder systems may exhibit various current-phase relations. Thus in this paper, we investigate the Josephson effects in systems consisting of *two* two-leg Kitaev ladders by using numerical diagonalization. Finding a variety of Josephson current-phase relationship would open up a wider range of the potential applications of the Kitaev chains other than topological quantum computation.

This paper is organized as follows. In section 2, we introduce the model Hamiltonian for the Josephson junction considered here. In section 3, we show the Josephson energy-phase relationship by using numerical diagonalization of the Hamiltonian. Finally, we summarize our results in section 4.

## 2 Model Hamiltonian

In this section, we introduce a junction system consists of the two Kitaev ladders. The Hamiltonian of the present system depicted in Fig. 1 is written in the standard notation as

$$H = H_{\mathrm{L}} + H_{\mathrm{R}} + H_{\mathrm{T}}, \tag{2}$$

$$H_{\mathrm{L}} = -\mu \sum_{i,j} c_{i,j}^{\mathrm{L}\dagger} c_{i,j}^{\mathrm{L}} - \frac{t_\perp}{2} \sum_i \left[ c_{i,1}^{\mathrm{L}\dagger} c_{i,2}^{\mathrm{L}} + c_{i,2}^{\mathrm{L}\dagger} c_{i,1}^{\mathrm{L}} \right]$$
$$- \frac{1}{2} \sum_{i,j} \left[ t\, c_{i+1,j}^{\mathrm{L}\dagger} c_{i,j}^{\mathrm{L}} + \Delta_j\, c_{i+1,j}^{\mathrm{L}\dagger} c_{i,j}^{\mathrm{L}\dagger} + \mathrm{h.c.} \right], \tag{3}$$

$$H_{\mathrm{R}} = -\mu \sum_{i,j} c_{i,j}^{\mathrm{R}\dagger} c_{i,j}^{\mathrm{R}} - \frac{t_\perp}{2} \sum_i \left[ c_{i,1}^{\mathrm{R}\dagger} c_{i,2}^{\mathrm{R}} + c_{i,2}^{\mathrm{R}\dagger} c_{i,1}^{\mathrm{R}} \right]$$
$$- \frac{1}{2} \sum_{i,j} \left[ t\, c_{i+1,j}^{\mathrm{R}\dagger} c_{i,j}^{\mathrm{R}} + \Delta_j \mathrm{e}^{i\theta}\, c_{i+1,j}^{\mathrm{R}\dagger} c_{i,j}^{\mathrm{R}\dagger} + \mathrm{h.c.} \right], \tag{4}$$

$$H_{\mathrm{T}} = -\frac{t_{\mathrm{T}}}{2} \sum_{j=1,2} \left( c_{N,j}^{\mathrm{L}\dagger} c_{1,j}^{\mathrm{R}} + c_{1,j}^{\mathrm{R}\dagger} c_{N,j}^{\mathrm{L}} \right), \tag{5}$$

where $c_{i,j}^{\mathrm{L}}/c_{i,j}^{\mathrm{R}}$ is the annihilation operator of the electron on the $i$-th site of the $j$-th chain to the left/right side of the junction. Thus the last contribution in (2), $H_{\mathrm{T}}$, corresponds to the tunnel Hamiltonian between the L/R ladders, given explicitly in (5). We note that the superconducting pair-potential $\Delta_j$ is only assumed within each chain. We also assume the two superconducting phase differences $\theta$ and $\phi$; $\theta$ is the Josephson phase difference across the junction, and $\phi$ is the phase difference between the chains $j = 1$ and 2. It is worth noting here that this interchain phase difference $\phi$ may induce several novel phenomena such as the modulation of the phase difference of the superconducting order parameter in two chains [8] or the increasing of crossed Andreev reflection [9]. Thus we introduce $\phi$ here in the hope that a novel phenomena would be induced again.

Throughout this paper, we take $t = 2\Delta = 1$ and $t_{\mathrm{T}} = 0.1$. The total number of sites per each chain is set to $N = 100$ for calculation of the Josephson effect. We compute the ground state energy of the whole system as a function of $\theta$ for several $t_\perp$, $\mu$, and also $\phi$ by using numerical diagonalization.

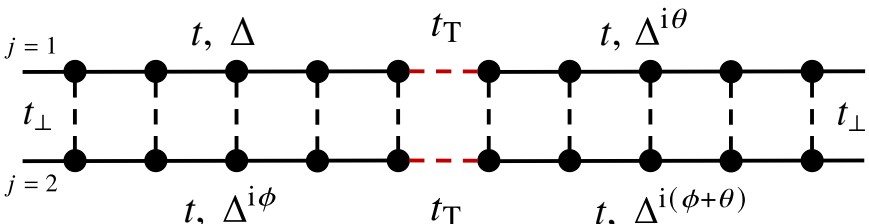

Figure 1: Schematic illustration of the junction system consisting of two Kitaev ladders. Note that the superconducting pair potential $\Delta$ is only assumed within each chain. Here, two superconducting phase differences are assumed; $\theta$ is the Josephson phase difference across the junction, and $\phi$ is the additional inter-chain phase difference.

## 3  Results

First, let us look at the Josephson energy-phase relationship $E_J(\theta)$ of the system (2). Figure 2 shows $E_J(\theta)$ for several values of $t_\perp/t$, fixing the other parameters as $\Delta/t = 0.5$, $\mu/t = 0.5$ and $\phi = \pi/3$. We find four qualitatively different types of behaviours of $E(\theta)$ as seen in Figs. 2 (a)-(d). Figure 2 (a) is exactly the topological Josephson energy-phase relationship with $4\pi$-periodicity. This is just because $t_\perp = 0$; there are two independent Josephson junctions of the Kitaev chains. In Fig. 2 (b), where $t_\perp/t = 0.1$, unusual behavior can be seen around $0.5 < \theta/\pi < 1.5$. The Josephson energy $E_J(\theta)$ exhibits a transition from an increasing function to a decreasing function of $\theta$ at around $\theta \simeq 0.5/\pi$. With further increasing $t_\perp/t$ to 0.3, $E_J(\theta)$ exhibits qualitatively different, but well-known $\theta$-dependence; the so-called $\pi$-junction as shown in Fig. 2 (c). Finally, if we take $t_\perp/t = 1.0$, the system switches back to the usual 0-junction property as shown in Fig. 2 (d).

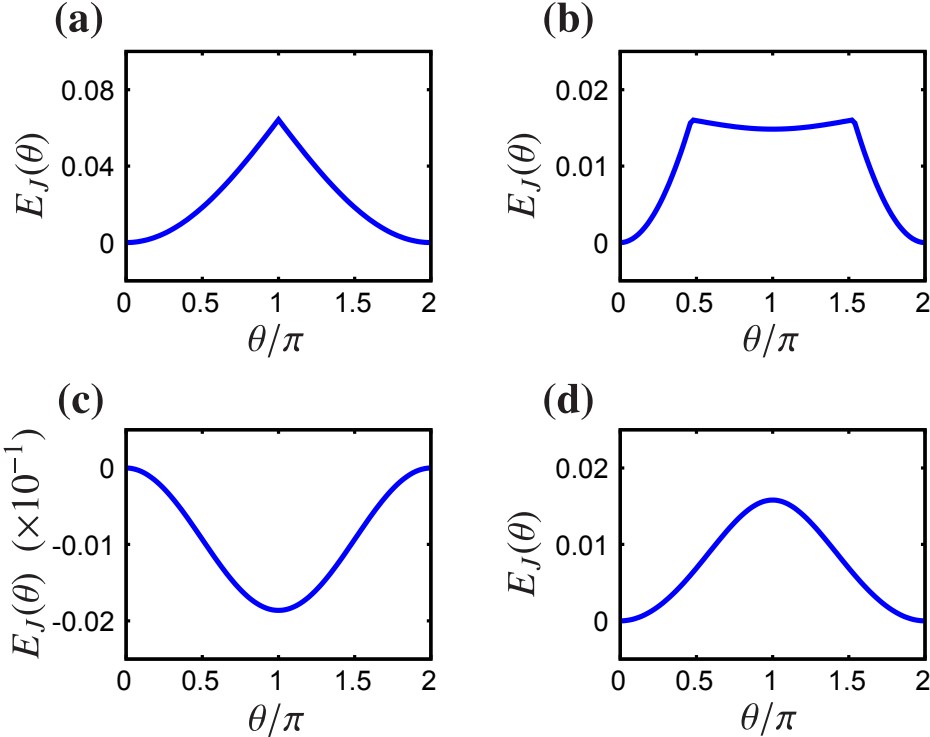

Figure 2: The Josephson energy-phase relation $E_J(\theta)$ of the ladder junction model when (a) $t_\perp/t = 0$, (b) $t_\perp/t = 0.1$, (c) $t_\perp/t = 0.3$, (d) $t_\perp/t = 1.0$.

Here, the question arises; how the four characteristic behaviors found here depend on the model parameters, in particular $\mu$ and $t_\perp$. Thus, we next look at the correlation between the types of $E_J(\theta)$ and the topological phases of the Kitaev ladder system. Before that, however, it is necessary to recall that the phase diagram of the ladder system is modified when $\phi > 0$ because the symmetry class of the system changes from BDI to D due to the broken time-reversal and chiral symmetries, and changes back to BDI when $\phi = \pi$ [10]. Thus, for $0 < \phi < \pi$, the system is characterized by a $\mathbb{Z}_2$ invariant $\mathcal{M}$ called the Majorana number, instead of the Chern number $w$ for $\phi = 0$ [7]. Following Maiellaro *et al.* [7], we calculate the Chern number $w$ and the Majorana number $\mathcal{M}$ for several values of $\phi$ to see the effects of the interchain phase difference $\phi$ on the topological phase diagram of the Kitaev ladder system, and the results are summarized in Fig. 3. For $\phi = 0$, the well-known phase diagram of the Kitaev ladder system

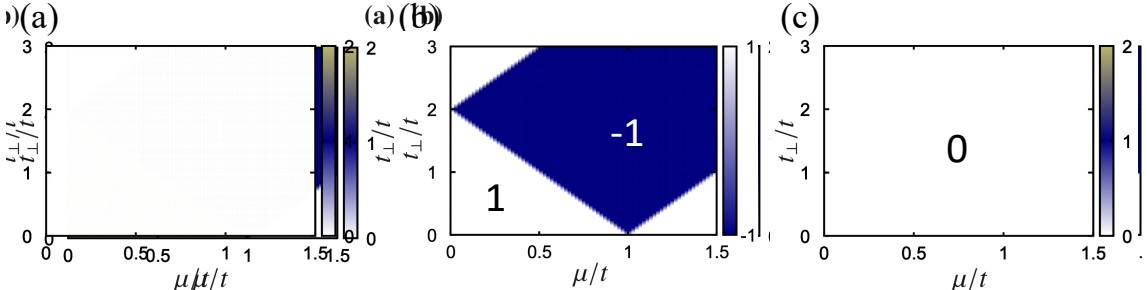

Figure 3: The topological phase diagrams of the Kitaev ladder system for (a) $\phi = 0$, (b) $\phi = \pi/3$, (c) $\phi = \pi$, by using the Chern number $w$ for (a) and (c), and also the Majorana number $\mathcal{M}$ for (b). The integers in the figures indicate the Chern number in (a) and (c), and the Majorana number in (b). The system is in the topologically nontrivial phases when $w \neq 0$ or $\mathcal{M} = -1$.

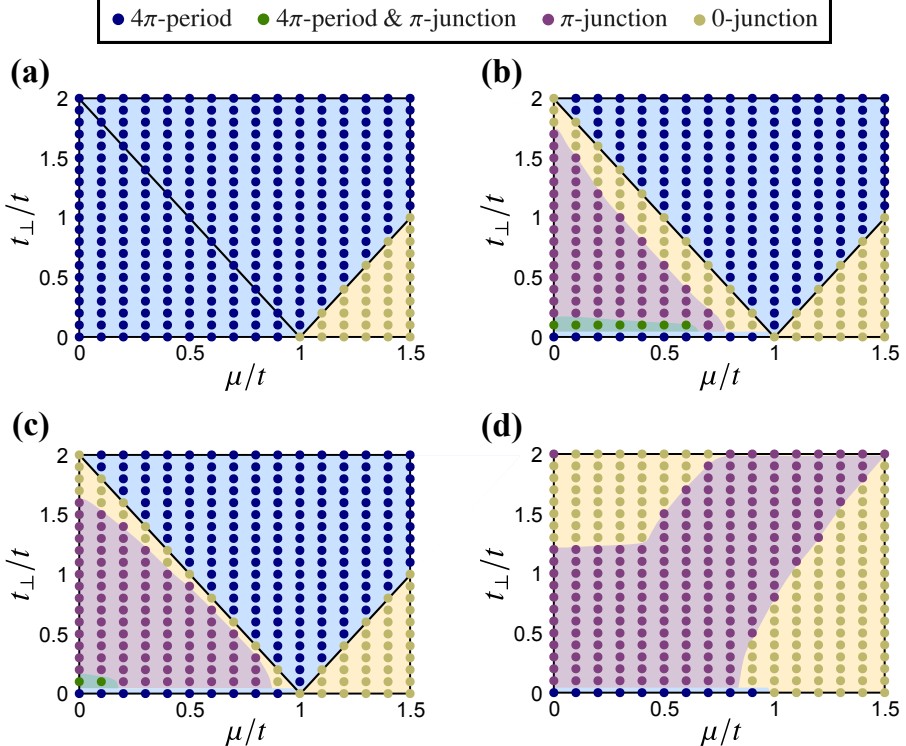

Figure 4: The correlation between the types of $E_J(\theta)$ on the phase diagram of the Kitaev ladder for (a) $\phi = 0$, (b) $\phi = \pi/3$, (c) $\phi = 2\pi/3$, and (d) $\phi = \pi$. Solid lines represent the boundary of the topological phase.

is reproduced as shown in Fig. 3 (a). We also confirm that the Chern number $w$ is 0 for any $\mu$ and $t_\perp$ when $\phi = \pi$, thus the system is always in a topologically trivial phase, as shown in Fig. 3 (c). If the symmetry class of the system is D when $0 < \phi < \pi$, the system is in a topologically nontrivial phase when $\mathcal{M} = -1$, and the region where this is the case shown in Fig. 3 (b) coincides with that with $w = 1$ when $\phi = 0$. Now we can see the relation between the types of Josephson coupling and the topological phases realized in the Kitaev ladder system based on the phase diagrams shown in Fig. 3.

Figure 4 shows which type of $E_J(\theta)$ appears on the phase diagram of the Kitaev ladder system calculated for several values of $\phi$. For $\phi = 0$ shown in Fig. 4 (a), we can see that the region where the topological Josephson energy-phase relationship with $4\pi$-periodicity (indicated by blue dots in the figure) appears coincides with the region with $w \neq 0$ in the phase diagram Fig. 3 (a). In the remaining region in Fig. 4 (a), where the $w = 0$, the system exhibits the 0-junction property. For $0 < \phi < \pi$ shown in Fig. 4 (b) and (c), the $\pi$-junction region appears (shown by red dots) in a fairly wide region of the phase diagram. We notice that the region where the $\pi$-junction appears almost coincides with the region with the Majorana number is 1 where the system is in the topologically *trivial* state. If we take $\phi = \pi$, the regions with the $4\pi$-junction disappears because the the Chern number $w$ is 0, i.e., there is no MZMs, in the whole region of the phase diagram, as shown in Fig. 3 (c). Instead, the $\pi$-junction region can be seen in wider range of the phase diagram. We should note here that the unusual $E_J(\theta)$ found in Fig. 2 is realized in the narrow region where $t_\perp/t$ is small shown in green dots in Fig. 4 (b) and (c). From this result, it is found that $4\pi$-period energy-phase relationship always appear in the region with $w \neq 0$, and $\pi$-junction appears in the region with $w = 0$ or $\mathcal{M} = 1$.More detailed studies on the origin of the various Josephson energy-phase relation $E_J(\theta)$ will be reported in a forthcoming paper.

## 4 Summary

In summary, we have investigated the Josephson energy-phase relationship in the Kitaev ladder systems by using numerical diagonalization, stimulated by the richer phase diagram and novel properties reported in the previous works [6–9]. We have found the variety of Josephson energy-phase relationships, such as the topological $4\pi$-, $\pi$-, and conventional 0-junctions. It is noteworthy that these Josephson current properties can be switched by externally controlling the interchain coupling $t_\perp$. This will open up new application possibilities [11–14] for the Kitaev superconducting chain other than topological quantum computation.

## Acknowledgements

Acknowledgements should follow immediately after the conclusion. Some numerical computations were carried out at the Cyberscience Center, Tohoku University, Japan.

**Funding information** This work was supported by JSPS KAKENHI Grant Number JP21H01025.

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
