# Peer review of "Josephson effects between the Kitaev ladder superconductors"

_SciPost Physics Proceedings_

## Round 1 · Referee Report · Anonymous (Referee 1) · 2023-2-14

Strengths

1- explicit results for observable manifestations of topological phases in a Kitaev ladder system 2- robust numerics 3- clear presentation of all findings.

Weaknesses

1- in the light of earlier works the findings are not too surprising, but they nicely illustrate how the various topological phases manifest themselves in the characteristics of the Josephson energies and currents.

Report

This manuscript reports various Josephson effects arising from a tunnel junction between two Kitaev ladders. The main results, presented in section 3, are the numerically determined ground state energies as a function of angles $\theta$ and $\phi$ representing superconducting phase differences between and across the ladders. The results are consistent with earlier findings of Maiellaro et al (cited as [7]) on the phase diagram of the Kitaev ladder.

Requested changes

1- p2: are also affect -> also affect 2- p2: Maiellaro et al. has shown -> Maiellaro et al. have shown 3- p4: while I understand how fig 2a represents the $4 \pi$ periodic case, it would be good to mention that the $4\pi$ period arises due to a crossing of two curves at $\theta=\pi$ (which as such is not visible in the figure) 4- p6: always appear -> always appears 5-p6: omit the sentence: Acknowledgements should follow immediately after the conclusion.

---

## Editorial Decision

resubmitted